# A transgenic toolkit for visualizing and perturbing microtubules reveals unexpected functions in the epidermis

Andrew Muroyama[1,3], Terry Lechler[1,3]*

[1]Department of Dermatology, Duke University Medical Center, Durham, United States; [3]Department of Cell Biology, Duke University Medical Center, Durham, United States

**Abstract** The physiological functions of microtubules (MTs) are poorly understood in many differentiated cell types. We developed a genetic toolkit to study MT dynamics and function in diverse cells. Using TRE-EB1-GFP mice, we found that MT dynamics are strongly suppressed in differentiated keratinocytes in two distinct steps due to alterations in both growth rate and lifetime. To understand the functions of these MT populations, we developed TRE-spastin mice to disrupt MTs in specific cell types. MT perturbation in post-mitotic keratinocytes had profound consequences on epidermal morphogenesis. We uncoupled cell-autonomous roles in cell flattening from non-cell-autonomous requirements for MTs in regulating proliferation, differentiation, and tissue architecture. This work uncovers physiological roles for MTs in epidermal development, and the tools described here will be broadly useful to study MT dynamics and functions in mammals.

DOI: https://doi.org/10.7554/eLife.29834.001

*For correspondence:
terry.lechler@duke.edu

**Competing interests:** The authors declare that no competing interests exist.

## Introduction

Many studies conducted over the last several decades have provided significant insight into microtubule-associated proteins (MAPs) and their effects on microtubule dynamics and organization in cultured cells. The functions for microtubules in intact tissues, as well as their organization and dynamics, are less well understood. There are multiple reasons for this: (a) many cell types will not fully differentiate when cultured outside of the organism, (b) non-specific effects resulting from using drug-induced microtubule disassembly assays, including secondary effects due to mitotic arrest and apoptosis, (c) lack of information about which microtubule regulators are essential for microtubule organization in many differentiated cell types, and (d) few genetic tools to specifically perturb microtubules in differentiated cell types. While differentiation induces loss of centrosomal microtubule-organizing center (MTOC) activity in many cells, there are only a few examples where both the changes in microtubule organization and dynamics and the functions of the resulting networks have been assayed in vivo (*Lacroix et al., 2014*; *Le Droguen et al., 2015*; *Oddoux et al., 2013*).

The mammalian epidermis develops from a single layer of proliferative basal cells that adheres to an underlying basement membrane. As stratification proceeds, cells transit outward and progressively differentiate, first into spinous cells, then into granular cells, and finally into corneocytes, which are required for epidermal barrier activity. While microtubules are organized in a centrosomal array in basal cells, differentiation results in loss of centrosomal MTOC activity, and MTs eventually organize into cortical arrays (*Lechler and Fuchs, 2007*; *Muroyama et al., 2016*). The functions for MTs in the differentiated cells remain unknown, although they have been implicated in the formation and function of cell-cell adhesions (*Nekrasova et al., 2011*; *Simard-Bisson et al., 2017*; *Sumigray et al., 2012*). Here, we characterize both the dynamics and the functions of epidermal microtubules using

novel transgenic mouse lines that reveal differentiation-induced alterations in MT dynamics as well as unexpected functional roles for non-centrosomal MT arrays in these cells.

## Results

### Visualizing microtubule dynamics during differentiation in vivo with the TRE-EB1 mouse

To visualize and quantify microtubule dynamics in vivo, we generated a transgenic mouse containing a cassette encoding a GFP-tagged copy of the microtubule plus-tip tracking protein EB1 under the tetracycline-responsive element (TRE) promoter (*Figure 1A*). With this line, EB1-GFP expression can be temporally controlled through doxycycline exposure and spatially controlled via cell- or tissue-specific tTA/rtTA lines (*Urlinger et al., 2000*) or tissue-specific Cre lines paired with the Rosa-rtTA line (*Hochedlinger et al., 2005*). After screening founders by assaying EB1-GFP induction in primary keratinocytes, we chose a line with mosaic expression (hereafter referred to as TRE-EB1) to allow precise resolution of single cells in a complex tissue field.

We generated CMV-rtTA; TRE-EB1 embryos to visualize microtubule dynamics in keratinocytes during progressive differentiation transitions with sub-cellular resolution (*Figure 1B*, *Figure 1—figure supplement 1A*). Importantly, we did not observe any cell morphology or tissue architecture phenotypes associated with EB1-GFP expression using this system. Microtubules in basal keratinocytes grew from the cell center to the periphery in a roughly radial orientation, in agreement with previous data that the centrosome is the primary MTOC in these cells (*Figure 1C*, *Video 1*) (*Lechler and Fuchs, 2007*). In spinous cells, clear radial organization was lost, and microtubules were observed growing in all directions (*Figure 1C*, *Video 2*). Despite the reorientation of microtubule growth in spinous cells, EB1 density was unaffected by this initial differentiation transition (*Figure 1—figure supplement 1B*). Similar to spinous cells, microtubules in granular cells grow throughout the cytoplasm in all directions (*Figure 1C*, *Video 3*). However, the density of EB1-GFP puncta was higher in granular cells, and projections over time revealed that microtubules in these cells exhibited greatly reduced dynamics (*Figure 1C,D*). Treatment of embryos with nocodazole eliminated the GFP puncta, strongly suggesting that these EB1-GFP comets mark plus-ends of very slowly growing and/or paused microtubules (*Figure 1—figure supplement 1C*).

Quantification of microtubule growth distances revealed a gradual decrease in motility over keratinocyte differentiation (*Figure 1E*, *Table 1*). Surprisingly, this gradual decrease in microtubule growth distance was due to two changes in microtubule behavior that occurred at distinct differentiation transitions. At the basal to spinous cell transition, microtubule growth speed was relatively unaffected, although there was a minor increase in growth speed in spinous cells (basal mean growth speed 11.1 ± 3.1 µm/min versus spinous mean growth speed 12.2 ± 3.3 µm/min) (*Figure 1F*, *Table 1*). These growth rates were similar to those observed in vivo in cells of the *C. elegans* egg-laying apparatus and in mouse muscle (*Lacroix et al., 2014*; *Oddoux et al., 2013*). In contrast, microtubule growth speeds were strongly suppressed as spinous cells differentiated into granular cells (granular mean growth speed 7.1 ± 3.7 µm/min) (*Figure 1F*, *Table 1*). Examination of the persistence of a single EB1-GFP puncta revealed that growth duration was significantly shorter in spinous versus basal keratinocytes (*Figure 1G*). The growth duration was unchanged between spinous and granular cells. The short growth periods likely reflect pause and/or catastrophe events that cannot be discriminated because EB1-GFP marks only growing microtubules. While there was no correlation between the growth speed and duration ($R^2$ = 0.002–0.19) or the growth speed and distance ($R^2$ = 0.004–0.2), there was a clear correlation between the length of time an EB1-GFP puncta moved and how far it traveled in basal ($R^2$ = 0.87) and spinous cells ($R^2$ = 0.85), as would be expected for microtubules polymerizing at a constant speed (*Figure 1—figure supplement 2*). Interestingly, this correlation is greatly reduced in granular cells ($R^2$ = 0.23), (*Figure 1—figure supplement 2*).

To assess the importance of quantifying microtubule dynamics in vivo, we compared the microtubule dynamics we observed during differentiation in intact embryos to dynamics in cultured primary keratinocytes. Primary keratinocytes begin to stratify when cultured, forming a proliferative basal layer and a single, post-mitotic, 'suprabasal' layer, which recapitulates some of the features of spinous cell differentiation (*Muroyama et al., 2016*). Absolute polymerization rates were dramatically

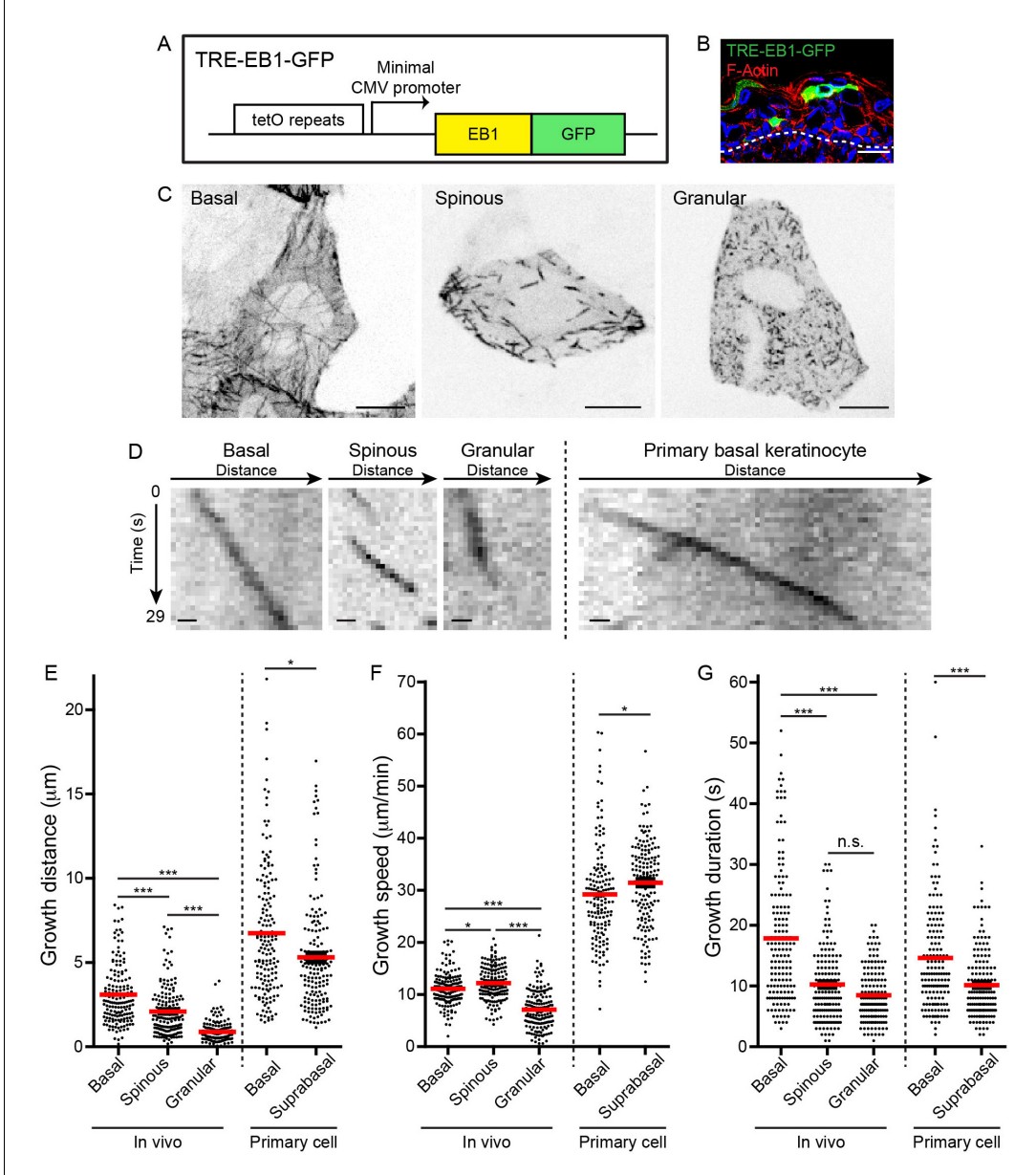

**Figure 1.** TRE-EB1 mouse line permits visualization of microtubule dynamics in vivo. (**A**) Diagram of the TRE-EB1-GFP transgene. (**B**) Cross-section of e17.5 CMV-rtTA; TRE-EB1 epidermis. Scale-20μm. (**C**) Representative standard deviation projections of a basal, spinous, and granular keratinocyte. Scale-10μm. (**D**) Kymographs of EB1-GFP in indicated cell types. Scale-1μm. (**E**) Quantification of microtubule growth distance. (**F**) Quantification of microtubule growth speed. (**G**) Quantification of duration of microtubule growth. n = 160 microtubules for each stage. Data are presented as mean ± S.E.M. n.s.-p>0.05, *p<0.05. ***p<0.001.

DOI: https://doi.org/10.7554/eLife.29834.005

The following figure supplements are available for figure 1:

**Figure supplement 1.** Validation of EB1-GFP line and characterization of microtubule density.
DOI: https://doi.org/10.7554/eLife.29834.006

**Figure supplement 2.** Correlation analysis of microtubule parameters for individual microtubules in the indicated cell types.
DOI: https://doi.org/10.7554/eLife.29834.007

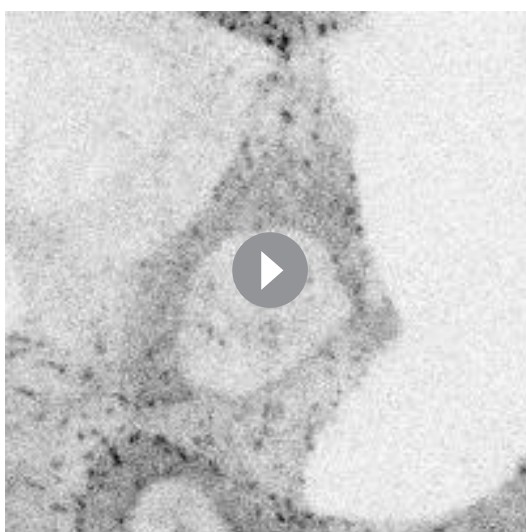

**Video 1.** EB1-GFP dynamics in a proliferative, basal keratinocyte in a mouse embryo.
DOI: https://doi.org/10.7554/eLife.29834.002

increased in isolated cells, suggesting that microtubule dynamics are altered as primary cells initiate a wound-healing response in culture, as has been noted for cell-cell junctions (*Figure 1D–G*) (*Foote et al., 2013*). Interestingly, although the absolute polymerization rate was greatly increased, the overall trends across all three measured parameters consistently mirrored the trends seen at the basal to spinous transition in vivo (*Figure 1E–G*). These data highlight that while there is some utility in assessing microtubule dynamics in cultured cells, they do not fully recapitulate the physiological setting.

Taken together, we have established the TRE-EB1-GFP mouse line as a tool to visualize and quantify microtubule behavior in single cells in vivo in mice. We used this line to make qualitative observations about microtubule organization during differentiation that confirmed that microtubules reorganize into non-centrosomal arrays as basal cells differentiate. Interestingly, by following microtubule dynamics over distinct differentiation transitions, we demonstrate that pause/catastrophe events are initially increased as basal cells differentiate into spinous cells. As spinous cells mature into granular cells, microtubule polymerization rates are strongly suppressed and the occurrence of very slow growing microtubules increases significantly. To determine the functions of these distinct populations of microtubules, we next created a transgenic tool to disrupt their organization.

## Development of TRE-spastin to genetically perturb MTs in vivo

Because loss-of-function approaches that specifically target differentiated cells are not always feasible in tissues that rapidly turn over, we pursued a gain-of-function strategy to disrupt microtubule organization via spastin overexpression (OE). Spastin is a single-subunit microtubule severing protein whose overexpression is sufficient to dramatically perturb microtubule organization in numerous cell types (*Le Droguen et al., 2015*; *Quintin et al.,*

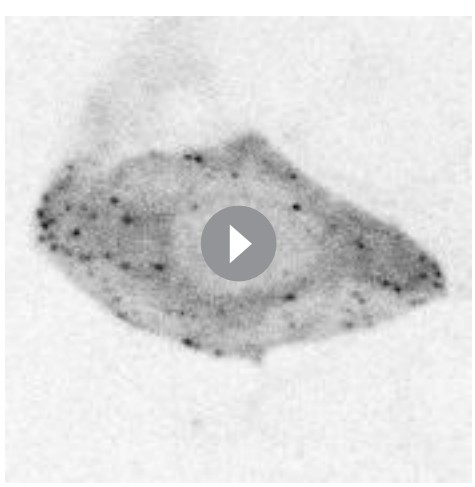

**Video 2.** EB1-GFP dynamics in a differentiated, spinous keratinocyte in a mouse embryo.
DOI: https://doi.org/10.7554/eLife.29834.003

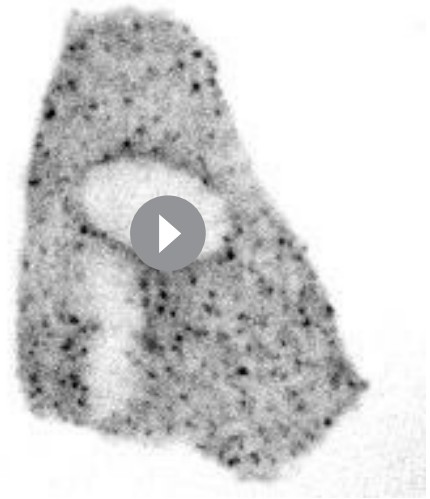

**Video 3.** EB1-GFP dynamics in a differentiated, granular keratinocyte in a mouse embryo.
DOI: https://doi.org/10.7554/eLife.29834.004

*2016*; *Sherwood et al., 2004*). Therefore, we placed the highly active M85 spastin isoform with an N-terminal HA tag under the TRE promoter, hereafter referred to as TRE-spastin (*Figure 2A*) (*Solowska et al., 2008*).

We validated the utility of this method in keratinocytes transfected with K14-rtTA and TRE-spastin plasmids. Spastin expression was only detected upon doxycycline administration, demonstrating that TRE-spastin can be temporally controlled in cultured cells. Importantly, all cells containing detectable spastin expression showed almost complete loss of microtubules, indicating that even low levels of spastin OE are sufficient to severely compromise microtubule organization (*Figure 2B–D*).

Next, we generated TRE-spastin transgenic mice to genetically perturb MT organization in vivo. To validate that spastin OE can (1) be temporally controlled and (2) disrupt microtubules in tissue, we globally induced spastin OE by administering doxycycline to CMV-rtTA; TRE-spastin mice. We detected robust spastin OE in CMV-rtTA; TRE-spastin mice within 24 hr of doxycycline administration, and we did not detect HA expression in doxycycline-treated TRE-spastin mice (*Figure 2E*). CMV-rtTA; TRE-spastin mice rapidly lost weight and had macroscopic alterations to some organs (*Figure 2F* and data not shown). Importantly, spastin OE greatly reduced α-tubulin signal in all tissues assayed; often only small fragments of MTs remained following spastin induction (*Figure 2G, H*). In addition to cytoplasmic microtubules, spastin OE also dramatically shortened cilia, although it is unclear if this treatment shortens existing cilia or perturbs new cilia growth (*Figure 2G*). Therefore, the TRE-spastin mouse can be used to genetically perturb microtubule organization in vivo to assess the functional roles for microtubules in numerous tissues.

## Microtubule disruption in proliferative cells of the mammalian epidermis

Next, we used the TRE-spastin mouse to understand how microtubules in distinct cell populations influence epidermal development. We generated K14-rtTA; TRE-spastin mice and induced spastin OE in proliferative basal keratinocytes (*Figure 3A*). Between 20–40% of basal cells overexpressed spastin using this strategy (*Figure 3B*). Spastin OE perturbed microtubule organization in basal cells and, as expected, induced a dramatic increase in mitotically arrested cells in mutant tissue (*Figure 3C–E*). Only spastin-positive cells in mutant tissue had uniformly unaligned, condensed chromosomes, demonstrating that spastin OE in basal keratinocytes caused a mitotic arrest in a cell-autonomous manner (*Figure 3F*). Consequently, we observed increased apoptosis, presumably due to mitotic catastrophe, and a tissue-wide hyper-proliferative response to maintain progenitor number (*Figure 3G,H*). At this level of induction, the tissue remained architecturally normal with no detectable barrier defects, demonstrating that the epidermis is highly robust to perturbations in basal keratinocytes (*Figure 3I*).

Interestingly, we noted spastin-positive cells in the upper layers of the epidermis after several days of induction in the K14-rtTA; TRE-spastin line. These suprabasal spastin-positive cells expressed keratin 5/14, which are normally exclusively expressed in basal keratinocytes (*Figure 3J*). As we have not detected any mis-expression of the K14-rtTA transgene, this result suggests that spastin-expressing cells delaminated from the basement membrane. Indeed, we identified many mitotically arrested cells that appeared to be in the process of delaminating, suggesting that spastin-positive cells in the suprabasal layers were generated, at least partially, through delamination (*Figure 3K*). Thus, the epidermis eliminates mitotically arrested cells from the proliferative niche either through apoptosis or delamination. Areas with numerous suprabasal spastin-positive cells displayed local thickening, suggesting that disruption of microtubules in suprabasal keratinocytes may more severely disrupt epidermal architecture (*Figure 3J*). Therefore, we next sought to specifically perturb microtubule organization in post-mitotic suprabasal cells, where their functions are unknown.

## Generation of K10-rtTA to specifically induce expression in suprabasal keratinocytes in vivo

Few tools currently exist to control transgene expression specifically in the suprabasal layers of the mammalian epidermis. Therefore, we generated a BAC transgenic in which rtTA is expressed from a large region of the keratin 10 promoter (*Figure 4—figure supplement 1A*). Using TRE-H2B-GFP mice, we validated that K10-rtTA faithfully recapitulates endogenous K10 expression in both

**Table 1.** Quantifications of microtubule parameters in indicated cell types.
Data are represented as mean ± standard deviation. n = 160 microtubules for each cell type.

| | | EB1 density (puncta/100 $\mu m^2$) | Mean growth distance ($\mu m$) | Mean growth speed ($\mu m/min$) | Mean growth duration (s) |
|---|---|---|---|---|---|
| In vivo | Basal | 9.1 ± 2.1 | 3.1 ± 1.77 | 11.09 ± 3.11 | 17.84 ± 10.98 |
| | Spinous | 8.68 ± 2.3 | 2.1 ± 1.5 | 12.2 ± 3.27 | 10.24 ± 6.32 |
| | Granular | 17.39 ± 4.64 | 0.89 ± 0.57 | 7.1 ± 3.66 | 8.49 ± 4.33 |
| Primary | Basal | 3.56 ± 1.17 | 6.74 ± 3.9 | 29.23 ± 9.19 | 14.63 ± 9.11 |
| | Suprabasal | 4.23 ± 1.76 | 5.31 ± 3.34 | 31.46 ± 7.97 | 10.15 ± 5.62 |

DOI: https://doi.org/10.7554/eLife.29834.008

embryos and adults (*Figure 4—figure supplement 1A*). K10-rtTA induction was observed in e14.5 epidermis, as stratification commences, and was robustly and uniformly induced by e15.5 (*Figure 4—figure supplement 1B*) (*Tumbar et al., 2004*). Robust expression was observed in all K10-expressing tissues assayed, with the exception of the embryonic dorsal tongue, which only exhibited minimal induction (*Figure 4—figure supplement 1C,D*). Importantly, H2B-GFP expression was only detected in cells that endogenously express K10, demonstrating that our K10-rtTA line is a powerful tool to reliably control expression exclusively in post-mitotic suprabasal cells, thereby bypassing any

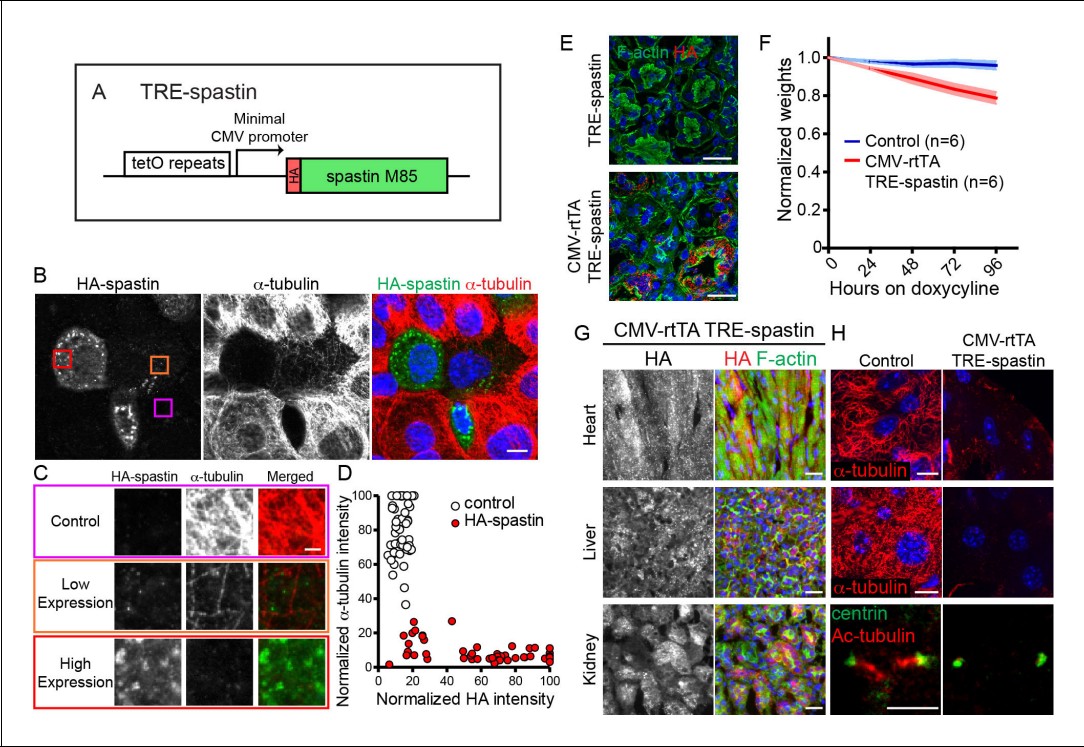

**Figure 2.** TRE-spastin expression perturbs microtubules in vitro and in vivo. (**A**) Diagram of the TRE-spastin transgene. (**B**) Spastin OE perturbs microtubules in cultured cells. Scale-10µm. (**C**) Insets from (**B**) showing microtubule density within individual cells based on spastin expression. Scale-2µm. (**D**) Quantification of microtubule perturbation following spastin OE. n = 50 cells each from two independent experiments. (**E**) Spastin expression was observed within 24 hr of doxycycline exposure and no leaky expression was detected in TRE-spastin mice. Scale-25µm. (**F**) Weights of CMV-rtTA; TRE-spastin mice and control littermates following doxycycline exposure. (**G**) HA-spastin expression in various tissues after 96 hr of doxycycline exposure. Scale-25µm. (**H**) Effects of spastin OE on microtubule density in vivo. Note that the cilia in the kidney are dramatically shortened. Scale for the heart and liver microtubules-10µm. Scale for the cilia-5µm.
DOI: https://doi.org/10.7554/eLife.29834.009

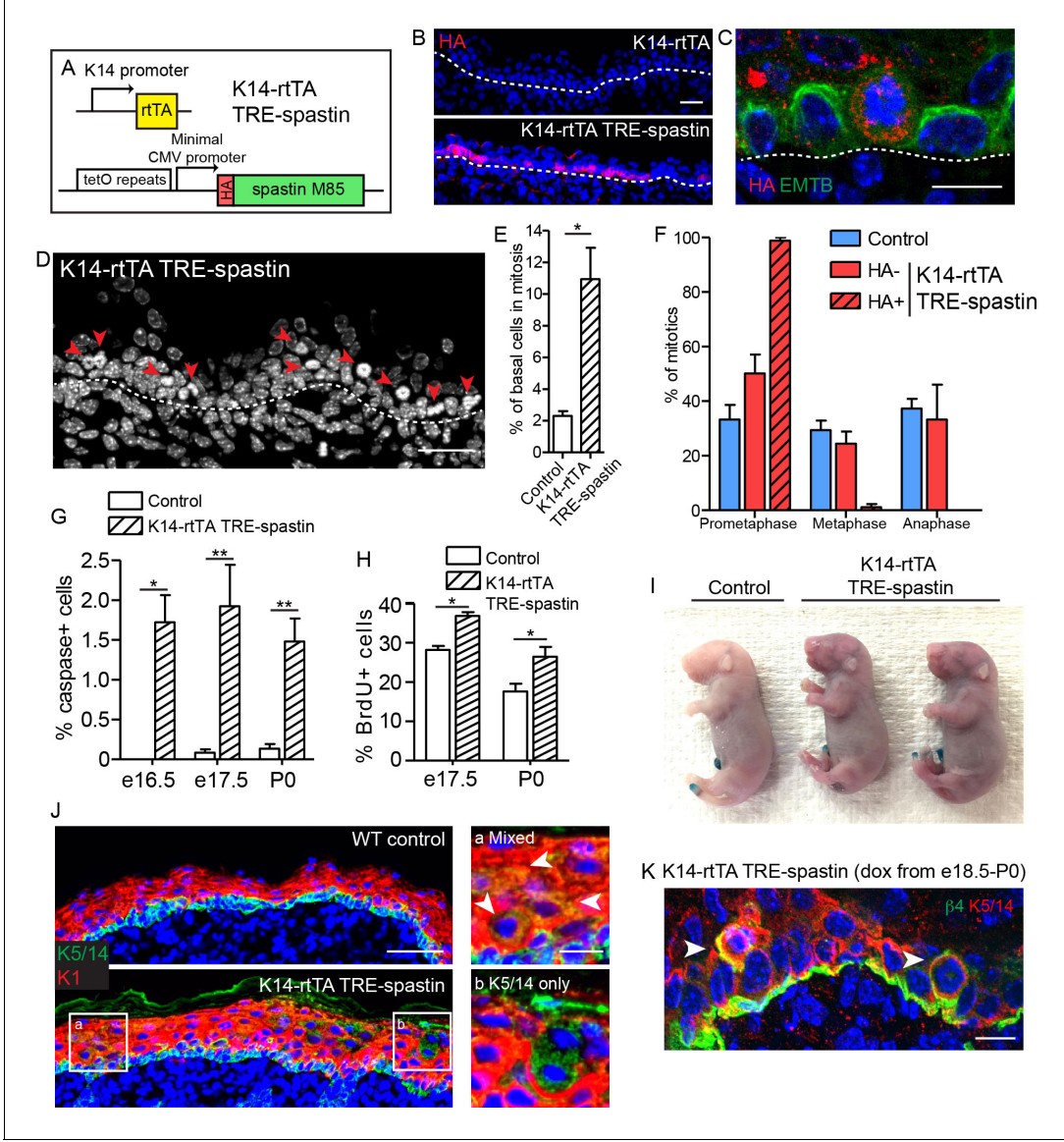

**Figure 3.** Spastin OE in basal keratinocytes induces mitotic arrest but does not alter epidermal architecture. (**A**) Alleles used to induce spastin overexpression in basal keratinocytes. (**B**) HA-spastin expression in e16.5 embryonic epidermis. Scale-25μm. (**C**) Spastin OE causes microtubule loss, assayed using the 3xGFP-ensconcin microtubule-binding domain (EMTB) mouse (*Lechler and Fuchs, 2007*), and mitotic arrest. Scale-10μm. (**D**) Arrows indicate mitotically arrested cells in K14-rtTA; TRE-spastin epidermis. Scale-25μm. (**E**) Quantification of the number of basal keratinocytes in mitosis. n = 3 mice per genotype. (**F**) Quantification of mitotic stage in control back skin and spastin-negative and spastin-positive cells in K14-rtTA; TRE-spastin back skin. n = 3 mice per genotype. (cell type x mitotic stage interaction, p<0.0001). (**G**) Quantification of cleaved-caspase-3-positive cells at the indicated stages. (**H**) Quantification of BrdU+ basal cells in control and mutant back skin at the indicated stages. (**I**) X-gal barrier assay in e18.5 embryos. (**J**) Expression of keratins 5/14 and keratin 1 in control and K14-rtTA; TRE-spastin epidermis. Scale-50μm. Insets show zoomed regions illustrating cells expressing both K5/14 and K1 (white arrows) and also suprabasal cells that are only K5/14+. Scale-10μm. (**K**) Delaminating mitotic cells in K14-rtTA; TRE-spastin epidermis. Scale-10μm. Data are presented as mean ± S.E.M. *p<0.05. **p<0.01.
DOI: https://doi.org/10.7554/eLife.29834.010

potential mitotic abnormalities that may result from perturbations utilizing the widely adopted K14-rtTA and K5/14-Cre systems.

## Disruption of MTs in differentiated cells induces epidermal hyperproliferation and profound architecture defects

To understand the functions of non-centrosomal MTs in the differentiated cells of the epidermis, we used K10-rtTA; TRE-spastin mice to overexpress spastin in post-mitotic spinous and granular cells from e16.5 (*Figure 4A*). We confirmed that spastin expression was confined to post-mitotic suprabasal keratinocytes by HA staining and found that spastin was expressed on average in 77% of suprabasal cells (*Figure 4B*). Mutant neonates were recognizable from control littermates, and some exhibited flaky skin (*Figure 4—figure supplement 2A*). Surprisingly, spastin OE in K10-rtTA; TRE-spastin mice led to a severe thickening of the epidermis and disruptions to epidermal architecture (*Figure 4C–E*), phenotypes not predicted by prior studies. Therefore, we concentrated on understanding the mechanisms by which MTs in suprabasal cells control tissue morphology.

We found that the increased thickness of the mutant epidermis was associated with an increase in the number of cell layers (*Figure 4E*). This increase in cell number was due to a specific hyperproliferation of basal progenitor cells, as no proliferation was noted in the suprabasal cell compartment (*Figure 4F*). These data demonstrate a non-cell-autonomous response of the progenitor cells to microtubule loss in their differentiated progeny. Hyperproliferation was not caused by elevated apoptosis, as apoptosis was not significantly increased (*Figure 4—figure supplement 2B,C*). Stratification occurred normally, as K5/14+ basal cells remained in a single layer above the basement membrane, but there was a dramatic thickening of the K10 +layers (*Figure 4—figure supplement 2D*). Taken together, our data indicate that spastin OE in differentiated keratinocytes causes dramatic hyperproliferation within the basal layer, leading to a severe thickening of all of the differentiated layers of the tissue. While barrier defects are known to induce compensatory proliferation of basal cells, we show below that the effects seen here are independent of barrier loss.

## Microtubules are required for differentiation-induced cell-shape changes

While hyperproliferation contributes to the epidermal thickening in mutant tissue, closer examination revealed that thickening was additionally driven by changes to cell shape. As keratinocytes transition from spinous to granular cells, they adopt a flattened shape that, when viewed in cross-section, is highly anisotropic (spinous mean aspect ratio (AR) = 1.88; granular mean AR = 5.08). Currently, little is known about how this cell-shape change is controlled. Strikingly, differentiating spastin-positive cells were incapable of properly flattening (spinous AR = 1.72 versus granular AR = 1.89) (*Figure 4G,H*). Cell-shape defects were cell-autonomous, as wild-type cells in K10-rtTA; TRE-spastin epidermis were still able to flatten, although were sometimes distorted by their spastin-positive neighbors. Identical cell-shape defects were also seen in granular cells expressing HA-spastin in K14-rtTA; TRE-spastin mice (*Figure 4—figure supplement 3*). These cell-shape changes were not due to gross loss of adherens junctions, as we noted no obvious disruptions to cortical E-cadherin localization in mutant tissue (*Figure 4—figure supplement 4*). This is consistent with E-cadherin loss-of-function mutants, which are still able to undergo squamous morphogenesis (*Tinkle et al., 2004*; *Tunggal et al., 2005*). Additionally, granular cell markers were still induced in the proper layers, indicating that inability to flatten is not due to a general block in differentiation (*Figure 4—figure supplement 2D*). These data reveal an unexpected role for microtubules in differentiation-induced cell flattening in the epidermis. Interestingly, these phenotypes were not observed upon deletion of type II myosins or core actin regulators such as the Arp2/3 complex (*Sumigray et al., 2012* ; *Zhou et al., 2013*), suggesting that changes in granular cell morphology are not driven by alterations in acto-myosin contractility and demonstrating a role for MTs in this process.

To address whether MTs are required for initial flattening and/or maintenance of flattening, we first performed short-term doxycycline treatment of K10-rtTA; TRE-spastin embryos (e18.5-P0) and measured the aspect ratio of granular cells, reasoning that many of the granular cells we observed in these back skins were flattened prior to spastin OE. Indeed, many flattened spastin-positive granular cells were observed in the upper granular layers in these mice (*Figure 4H,I*). Additionally, we treated isolated granular cells with nocodazole and noted no change in cell morphology (*Figure 4J*). Therefore, microtubules are required for the cell flattening at the spinous to granular transition but are dispensable for maintaining the established flattened shape.

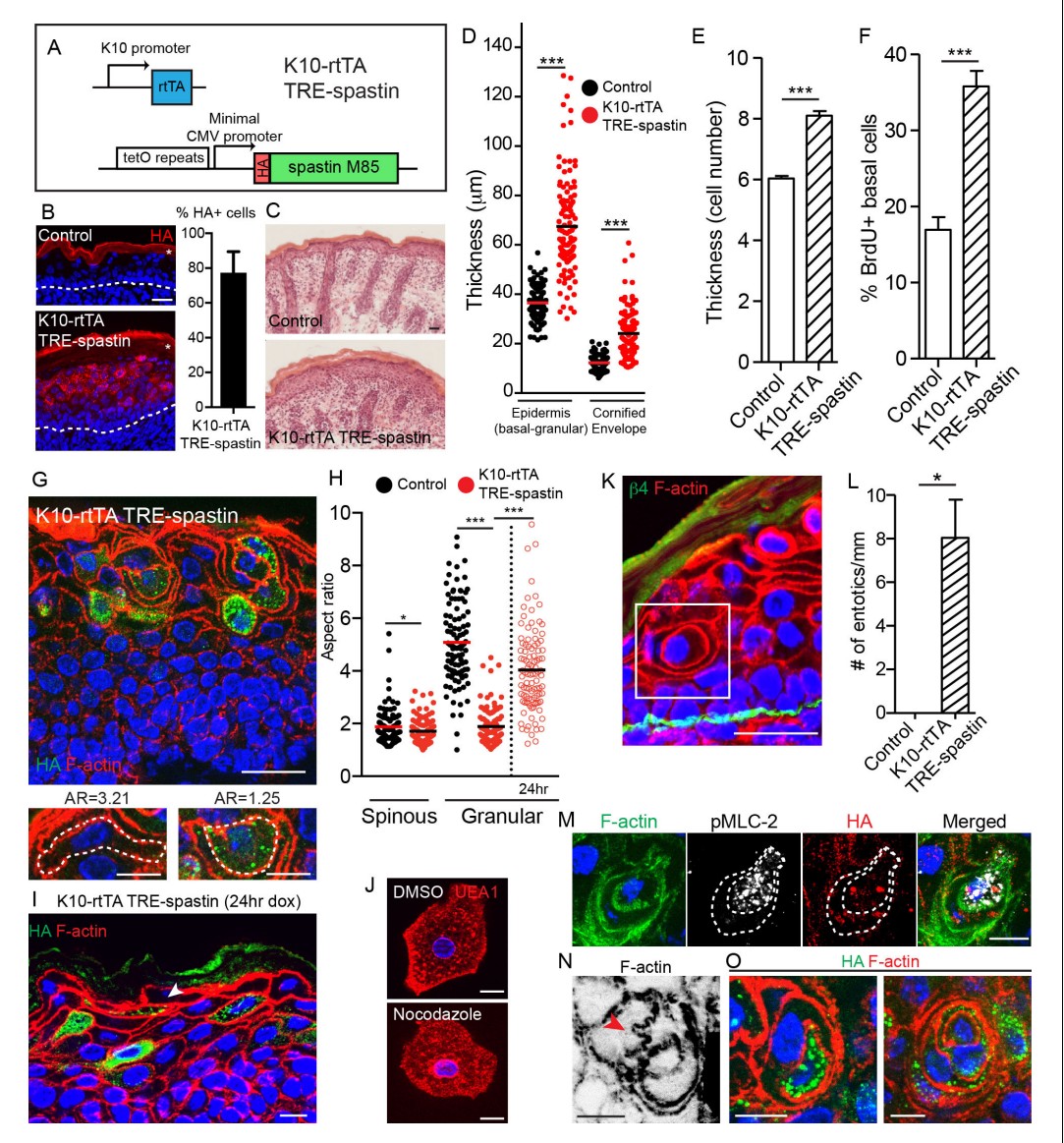

**Figure 4.** Spastin OE in differentiated keratinocytes induces cell-shape changes and entosis. (A) Alleles used to overexpress spastin in suprabasal keratinocytes. (B) Images and quantification of HA-spastin expression in control and K10-rtTA; TRE-spastin epidermis. Asterisks indicate autofluorescence of the cornified envelope. Scale-25μm. Quantification is the percentage of suprabasal cells expressing spastin averaged over 5 different mice. (C) Hematoxylin and eosin staining of control and K10-rtTA; TRE-spastin tissue. Note the cornified envelope thickness. Scale-25μm. (D) Quantification of epidermal thickness in control and K10-rtTA; TRE-spastin mice. Each column is 120 measurements from 4 mice per genotype. (E) Quantification of the number of cell layers present in control and K10-rtTA; TRE-spastin epidermis. n = 100 measurements from 4 mice per genotype. (F) Quantification of BrdU+ basal cells in control and K10-rtTA; TRE-spastin epidermis. n = 4 mice per genotype. (G) Cell rounding is observed in a cell-autonomous manner in K10-rtTA; TRE-spastin tissue. Scale-25μm. Zoomed regions show a spastin-negative and a spastin-positive cell within K10-rtTA; TRE-spastin tissue. Note the accompanying aspect ratios (AR). Scale-10μm. (H) Quantification of the aspect ratio of individual control and spastin-positive cells. n = 100 cells for each group. (I). Spastin-positive granular cells remain flattened after short spastin induction. Scale-10μm. (J) Isolated granular cells treated with DMSO or nocodazole. Scale-10μm. (K) Example of an entotic cell in K10-rtTA; TRE-spastin epidermis. Scale-25μm. (L) Quantification of the number of entotics per mm of basement membrane. n = 4 mice per genotype. (M) Example of an entosis where the invading cell has up-regulated phospho-myosin light chain II. The dotted line marks the cell outlines. Scale-10μm. (N) Example of cell potentially invading its neighbor. Scale-10μm. (O) Examples of types of entosis observed in K10-rtTA; TRE-spastin epidermis. Scale-10μm. Data are presented as mean ± S.E.M. *p<0.05, ***p<0.001.

DOI: https://doi.org/10.7554/eLife.29834.011

The following figure supplements are available for figure 4:

**Figure supplement 1.** K10-rtTA expression faithfully recapitulates endogenous K10 expression.

*Figure 4 continued on next page*

*Figure 4 continued*

DOI: https://doi.org/10.7554/eLife.29834.012

**Figure supplement 2.** Characterization of K10-rtTA; TRE-spastin epidermis.

DOI: https://doi.org/10.7554/eLife.29834.013

**Figure supplement 3.** Granular cells fail to flatten in the K14-rtTA; TRE-spastin epidermis.

DOI: https://doi.org/10.7554/eLife.29834.014

**Figure supplement 4.** Localization of E-cadherin in control and K10-rtTA; TRE-spastin epidermis.

DOI: https://doi.org/10.7554/eLife.29834.015

**Figure supplement 5.** Spastin OE in suprabasal keratinocytes in adult mice perturbs epidermal homeostasis.

DOI: https://doi.org/10.7554/eLife.29834.016

While adherens junctions appeared normal in the mutant epidermis, we were also interested in whether desmosomes, which are known upstream regulators of microtubule organization (*Lechler and Fuchs, 2007*; *Sumigray et al., 2011*), were perturbed and could underlie the cell-shape defects observed in K10-rtTA; TRE-spastin suprabasal cells. While some data in cultured cells suggests MTs are required for desmosome assembly, other studies have found little to no effect of MT disruption on desmosome formation in culture (*Nekrasova et al., 2011*; *Pasdar et al., 1992*; *Simard-Bisson et al., 2017*; *Sumigray et al., 2011*). We found a significant loss of cortical desmosomal protein staining (including DSG1, desmoplakin, and DSC2/3) in the spastin OE epidermis (*Figure 5A–C*). Ultrastructural analysis confirmed that desmosomes were smaller in K10-rtTA; TRE-spastin epidermis compared to control tissue (*Figure 5D*). Surprisingly, however, loss of cortical desmosomal components was non-cell autonomous; line-scan analysis confirmed that cortical localization of desmoplakin and DSC2/3 were similarly disrupted between two spastin-positive, one spastin-positive and one control, and two control cells in K10-rtTA; TRE-spastin epidermis (*Figure 5B,C*).

To determine whether microtubule disruption could intrinsically influence desmosome assembly, we examined pairs of spastin-positive cells surrounded by wild-type neighbors in K14-rtTA; TRE-spastin mice, where the percentage of spastin-expressing cells was lower. Desmoplakin localization was normal between these single spastin OE pairs within otherwise wild-type cells, arguing that microtubule disruption does not autonomously alter desmosome assembly in the epidermis (*Figure 5E,F*). However, in areas where a significant number of cells expressed HA-spastin, we saw the same desmosomal phenotypes as those observed in the K10-rtTA; TRE-spastin line (*Figure 5—figure supplement 1A,B*). Similarly, desmoglein-1 was also maintained at cell borders of single spastin expressing cells in the K14-rtTA; TRE-spastin line and was globally lost in areas where many suprabasal cells expressed spastin (*Figure 5—figure supplement 1C,D*). Therefore, the observed desmosome defects are not a primary effect of microtubule disruption, but rather are a tissue-wide response to it. Examination of spastin-positive cell pairs also clearly demonstrated that the cell-shape changes are not secondary to desmosome dysfunction, as these spastin OE cells with normal cortical desmoplakin levels also fail to flatten (*Figure 5E*). Finally, because basal cell hyperproliferation is not seen in desmoplakin-mutant skin, the progenitor hyperproliferation we observe in K10-rtTA; TRE-spastin epidermis is not likely to be secondary to desmosome disruption.

Strikingly, microtubule disruption resulted in a remarkable number of entosis-like structures in both the spinous and granular layers (*Figure 4K,L*). Imaging whole-cell volumes provided further evidence that the cell-in-cell structures strongly resemble entotic events (*Video 4*). Canonical entosis is an active invasion of one cell into the cytoplasm of another as it loses substrate attachment (*Overholtzer et al., 2007*). We observed invading cells with elevated pMLC-2, consistent with an active actin-based entotic invasion (*Figure 4M*). Additionally, we observed F-actin based protrusions that appeared to be invading into neighboring cells, potentially reflecting the initial stages of entosis (*Figure 4N*). Entotic events ranged from entosis of a single spastin+ cell to what appeared to be concentric rings of cells, suggesting multiple layers of entosis (*Figure 4O*). We did not observe any examples of a wild-type cell within a spastin+ cell. Taken together, our data demonstrate that microtubules prevent entosis in vivo, although the underlying mechanism remains unknown. One possibility is that without microtubules, differentiated keratinocytes cannot flatten and consequently invade one another, potentially due to imbalances in membrane tension. Other possibilities, such as a stress and/or transcriptional responses, may also underlie this phenomenon. This phenotype was not noted

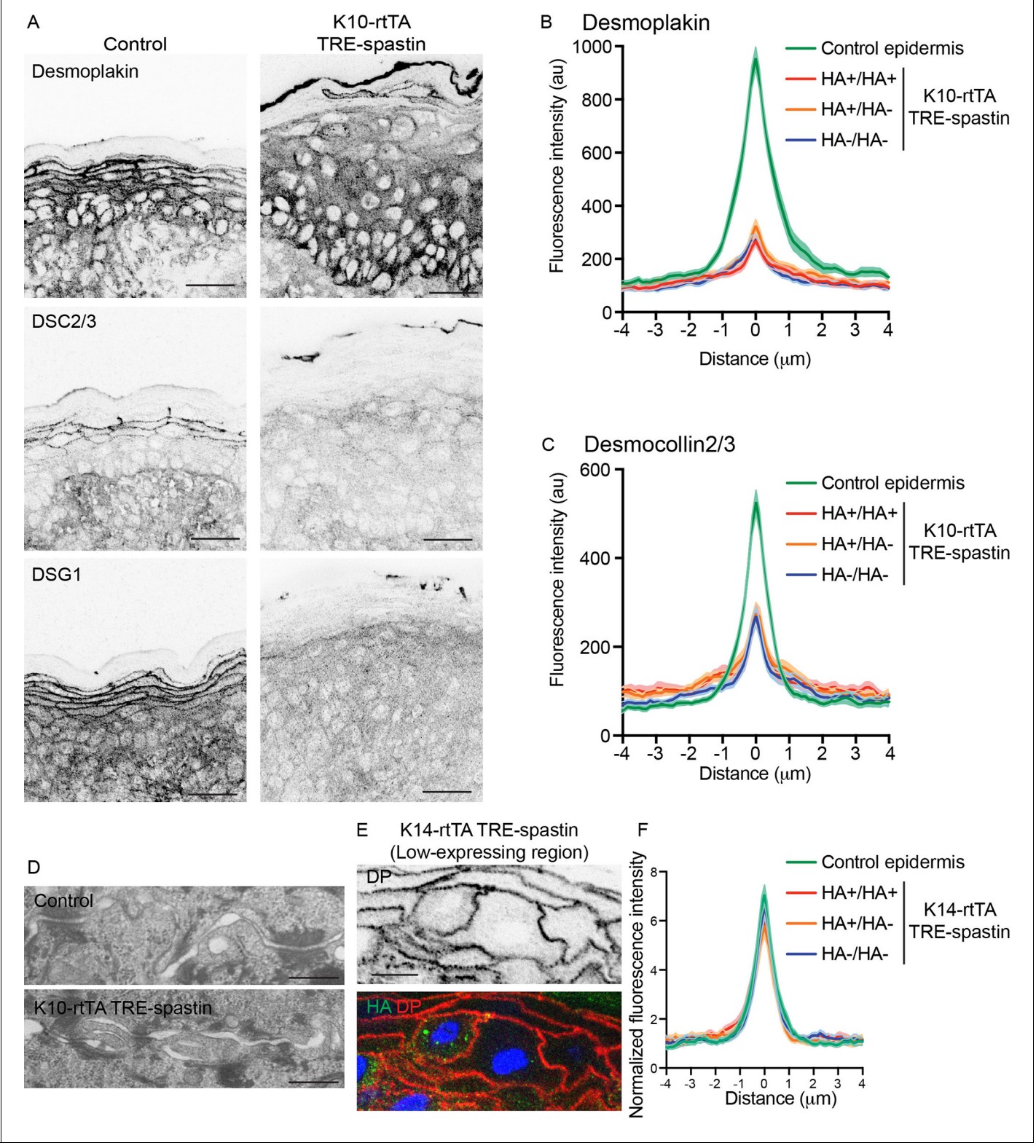

**Figure 5.** Non-cell-autonomous desmosome defects in K10-rtTA; TRE-spastin epidermis. (**A**) Immunofluorescence of desmosome components in control and K10-rtTA; TRE-spastin epidermis. Scale-25μm. (**B,C**) Quantifications of desmoplakin and DSC2/3 immunofluorescence at cell-cell boundaries between indicated cell pairs in K10-rtTA; TRE-spastin epidermis. n = 40 pairs from 2 mice for each pair type. (**D**) Transmission electron micrographs of desmosomes in control and K10-rtTA; TRE-spastin epidermis. Scale-500nm. (**E**) A pair of spastin-positive cells in K14-rtTA; TRE-spastin epidermis showing that spastin expression does not intrinsically alter cortical desmoplakin localization. Scale-10μm. (**F**) Quantification of desmoplakin

*Figure 5 continued on next page*

*Figure 5 continued*

immunofluorescence at cell-cell boundaries between indicated cell pairs in K14-rtTA; TRE-spastin epidermis with sparse HA-spastin suprabasal cells. Control (n = 25 pairs), HA+/HA+ (n = 15 pairs), HA+/HA- (n = 26 pairs), and HA-/HA- (n = 36 pairs) from 2 mice for each pair type.

DOI: https://doi.org/10.7554/eLife.29834.017

The following figure supplement is available for figure 5:

**Figure supplement 1.** Characterization of desmosomes in K14-rtTA; TRE-spastin epidermis.

DOI: https://doi.org/10.7554/eLife.29834.018

in desmoplakin-null epidermis (*Vasioukhin et al., 2001*), again suggesting that this is not secondary phenotype associated with adhesion defects.

The phenotypes described above – thickening of the epidermis and cell-shape defects – were also found upon disruption of microtubules in the adult epidermis, demonstrating that these phenotypes are not specific to embryonic development. Rather, microtubules are similarly essential for the homeostasis of the epidermis throughout life (*Figure 4—figure supplement 5*).

## Non-centrosomal microtubules are required for proper corneocyte formation but are dispensable for barrier function

Next, we wanted to assess whether the profound disruptions to epidermal architecture upon microtubule loss resulted in impaired barrier function. Epidermal barrier function is conferred through both tight junctions, which form in the granular layer, and the cornified envelopes, which are composed of enucleated, highly cross-linked corneocytes. Immunostaining for the tight junction proteins ZO-1 and occludin did not reveal any defects in spastin-positive cells or K10-rtTA; TRE-spastin tissue, demonstrating that microtubules are not required for localization of tight junction proteins in the mammalian epidermis (*Figure 6—figure supplement 1A–C*). Additionally, tight junctions halted biotin diffusion in mutant tissue, demonstrating that tight junction function is not observably impaired upon microtubule disruption (*Figure 6—figure supplement 1D*).

We noted that microtubule perturbation in K10-rtTA; TRE-spastin embryos caused formation of an abnormally thick cornified envelope (*Figure 6A*). By TEM, the thickened mutant CE was electron dense and many cytoplasmic remnants were observed, demonstrating a defect in corneocyte formation (*Figure 6B*). We confirmed that a subset of cytoplasmic proteins was retained in corneocytes in mutant back skin (*Figure 6C*). Retention of cytoplasmic components appeared to be cell autonomous, as we observed clear co-localization of the irregular staining with spastin-positive corneocytes (*Figure 6D*). Isolation of the cornified envelopes confirmed that spastin OE caused severe defects in corneocyte morphology (*Figure 6E,F*).

Because of the numerous cornified envelope defects, we performed a barrier assay to test epidermal exclusion of X-gal. Surprisingly, despite the numerous morphological defects in corneocytes, K10-rtTA; TRE-spastin embryos formed a fully functional barrier by e18.5, consistent with the fact that these mice survive postnatally (*Figure 6G*). We speculate that the thickening of the stratum corneum compensates for abnormal corneocyte formation. That said, our data reveal a previously unrecognized role for microtubules in the proper formation of the cornified envelope, which could either be a direct effect on corneocyte assembly or secondary to cell shape defects in granular cells.

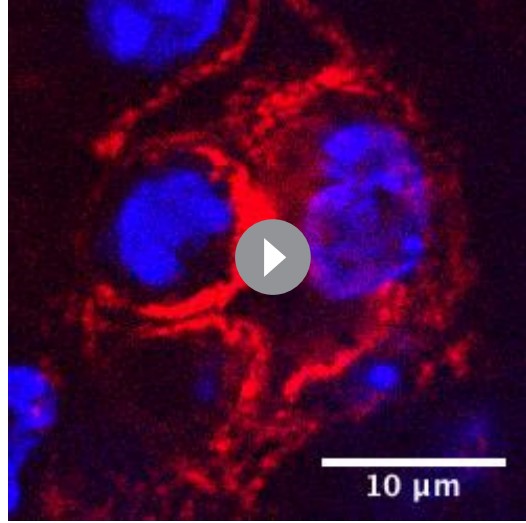

**Video 4.** Z-stack of an example of entosis in K10-rtTA; TRE-spastin epidermis. Phalloidin marks the cell outlines.

DOI: https://doi.org/10.7554/eLife.29834.019

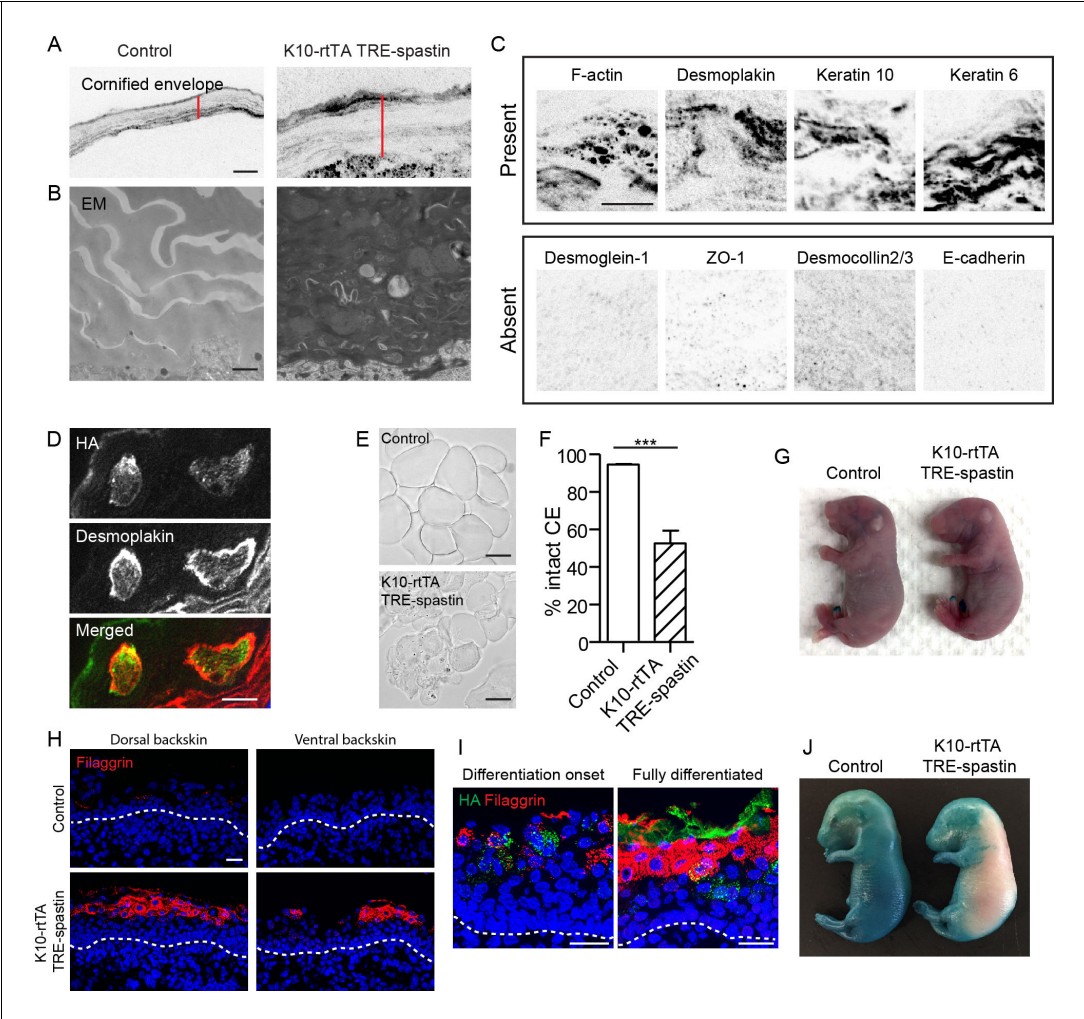

**Figure 6.** Proper corneocyte formation requires microtubules, but microtubule loss does not impair epidermal barrier function. (**A**) CE is thickened in K10-rtTA; TRE-spastin tissue. Red lines indicate CE thickness. Scale-10μm. (**B**) Transmission electron micrographs of cornified envelopes in control and K10-rtTA; TRE-spastin epidermis. Scale-500nm. (**C**) Examples of protein localization in the corneocytes of K10-rtTA; TRE-spastin mice. All images are inverted fluorescence (black indicates signal). All of the indicated proteins are absent in wild-type corneocytes. Scale-10μm. (**D**) Spastin expression cell-autonomously causes abnormal retention of cytoplasmic proteins. Scale-10μm. (**E**) Isolated corneocytes from control and K10-rtTA; TRE-spastin mice. Scale-25μm. (**F**) Quantification of isolated cornified envelopes. n = 40 random fields from 4 mice for each genotype. (**G**) X-gal barrier assay on e18.5 control and K10-rtTA; TRE-spastin embryos. (**H**) Epidermal cross-sections from e16.5 control and K10-rtTA TRE-spastin embryos, stained for the differentiation marker filaggrin. Scale-25μm. (**I**) Premature differentiation is non-cell autonomous in K10-rtTA; TRE-spastin epidermis. Filaggrin is induced in both spastin-positive and spastin-negative cells in prematurely differentiating K10-rtTA; TRE-spastin epidermis. Spastin-positive cells in the spinous layer do not induce filaggrin. Scale-25μm. (**J**) K10-rtTA; TRE-spastin e16.5 embryos prematurely form an epidermal barrier. Data are presented as mean ± S.E.M. ***p<0.001.

DOI: https://doi.org/10.7554/eLife.29834.020

The following figure supplement is available for figure 6:

**Figure supplement 1.** Spastin OE does not impair tight junction localization or function.

DOI: https://doi.org/10.7554/eLife.29834.021

## Microtubule disruption causes premature epidermal differentiation and barrier formation

Having identified cell-autonomous and non-cell-autonomous functions for microtubules late in embryonic development, we turned to understand how microtubules influence the earlier stages of epidermal differentiation. We analyzed K10-rtTA; TRE-spastin embryos where spastin induction began at e14.5 as stratification commences. Surprisingly, the epidermis in mutant embryos was

severely thickened even by e16.5, demonstrating that basal-cell hyperproliferation is a rapid response to suprabasal microtubule disruption, even before the barrier normally forms (*Figure 6H*). Concurrent with thickening, mutant epidermis exhibited signs of premature differentiation, including robust expression of the granular marker filaggrin, ZO-1, and the presence of cornified envelopes (*Figure 6H* and data not shown). Premature differentiation was not confined to spastin-positive cells; wild-type cells in e16.5 K10-rtTA; TRE-spastin epidermis also expressed filaggrin in the upper layers of the tissue (*Figure 6I*). We performed the X-gal exclusion assay on e16.5 embryos to determine if premature differentiation in K10-rtTA; TRE-embryos resulted in premature barrier formation. Interestingly, while control embryos had no barrier function at this stage, the back skin of K10-rtTA; TRE-spastin embryos had already formed a functional barrier (*Figure 6J*). Therefore, microtubule disruption in suprabasal cells during early epidermal stratification unexpectedly induces premature differentiation and barrier formation, potentially through dramatic tissue thickening.

## Discussion

We generated a TRE-EB1-GFP mouse line, which permits in vivo imaging of microtubule dynamics in various cell types. Using this mouse, we performed live imaging of EB1-GFP in the developing epidermis of intact mouse embryos and demonstrate that microtubule growth is suppressed as keratinocytes differentiate. To our knowledge, only one other quantification of MT dynamics over differentiation in an intact tissue—the *C. elegans* egg laying apparatus—has been performed (*Lacroix et al., 2014*). Interestingly, specific growth parameters were tuned at distinct transitions during keratinocyte differentiation. As basal cells differentiate into spinous cells, pause/catastrophe frequency was increased, as inferred by a decrease in EB1-GFP persistence. As spinous cells matured, polymerization rates were strongly suppressed. These data raise the intriguing possibility that specific subsets of MAPs that underlie these dynamic differences may be expressed at distinct stages of keratinocyte differentiation. Mining of published transcriptional databases reveals that epidermal differentiation induces extensive alterations to the expression of microtubule-related proteins, including tubulin isoforms ($\alpha,\beta,\gamma$), MAPs, and microtubule-modifying enzymes. A clear area of future work will be to determine which MAPs are responsible for the observed changes in microtubule growth parameters, with a special focus on determining how specifically tuned microtubule dynamics influence epidermal development. Furthermore, we anticipate that the TRE-EB1 mouse will be useful for performing similar measurements in vivo across differentiation lineages in other tissues.

We developed the TRE-spastin mouse to understand the functions for microtubules in diverse and distinct cell populations in vivo. While care must be taken to validate that spastin overexpression disrupts microtubules in other tissues of interest (*Sherwood et al., 2004*), we have found robust loss of microtubules in epidermis, intestine, heart and liver upon spastin induction using this line. This provides both the first experimental examination of microtubule function in intact epidermis and provides a microtubule 'null' phenotype that is essential for future comparisons to perturbations that affect microtubule organization and/or dynamics. The dramatic consequences of microtubule disruption in the epidermis suggest that more subtle changes to dynamics/organization may have phenotypic consequences. Ablation of microtubules in different epidermal compartments revealed distinct requirements for microtubules in epidermal morphogenesis. We found that the epidermis eliminates mitotically arrested cells through both apoptosis and delamination but is quite robust in responding to microtubule defects in a substantial number of cells. The elimination of basal cells by delamination/differentiation allows the epidermis to make use of these defective progenitors.

By overexpressing spastin in suprabasal keratinocytes using a novel K10-rtTA mouse, we demonstrate that stable non-centrosomal microtubules regulate overall tissue architecture. While there are pleiotropic phenotypes downstream of microtubule loss in differentiated keratinocytes, defects in cell shape appear to be one clear cell-autonomous consequence of microtubule disruption. In the absence of MTs, epidermal cells do not undergo squamous morphogenesis, the mechanisms of which remain poorly understood. Differentiation-induced flattening occurs in the absence of E-cadherin, type II myosins and the Arp2/3 complex, which clearly impact cell shape in numerous other cell types (*Sumigray et al., 2012*; *Tinkle et al., 2004*; *Tunggal et al., 2005*). How could microtubules promote cell flattening? One possibility is that microtubules influence cell shape indirectly through effects on other cytoskeletal elements, such as actin filaments, or on cell adhesions, such as

desmosomes. However, our data strongly argue against the latter as the flattening defects we observed were independent of any gross desmosome defects; single spastin-positive cells with normal cortical desmoplakin failed to flatten. An intriguing possibility is that kinesin-dependent microtubule sliding may provide a force to drive cell-shape changes, similar to what has been reported in neurons (*Winding et al., 2016*). Interestingly, our data also indicate that, while microtubules are required for initiating flattening, they are dispensable for maintaining the flattened granular shape, which could be controlled by other cytoskeletal elements like keratins.

One of the clear strengths of the TRE-spastin mouse is the ability to distinguish between cell-autonomous and tissue-wide effects of microtubule disruption. Spastin OE in suprabasal keratinocytes uncovered numerous unexpected non-cell-autonomous defects associated with loss of microtubules in differentiated cells. These included desmosome perturbation, basal hyperproliferation, cornified envelope defects, and premature differentiation. The mechanisms underlying these effects are a clear area for future study. Particularly, it remains important to identify how cells sense microtubule loss, whether the signal is mechanical in nature or more directly involves sensing tubulin/microtubule levels or organization. The work presented here highlights the value of examining phenotypes associated with microtubule disruption in vivo, and these tools should be of broad utility to the study of microtubules during mammalian development and homeostasis.

## Materials and methods

### Mice and tissues

All mice were maintained in accordance with Duke IACUC-approved protocols. To generate the TRE-EB1-GFP transgenic mouse line, EB1-GFP was digested out of K14-EB1-GFP with SacII and NotI and ligated into pTre2 cut with the same (*Muroyama et al., 2016*). *MAPRE1* is the official name for the gene encoding Eb1. The XhoI site next to the SapI site in pTre2 was mutated using site-directed mutagenesis (pTre XhoI mut). Proper doxycycline-dependent expression of the TRE-EB1-GFP vector was verified in cultured keratinocytes co-transfected with a K14-rtTA plasmid and placed in doxycycline-containing media for 16 hr. TRE-EB1-GFP was linearized using XhoI and was used by the Duke Transgenic Core to generate transgenics via pronuclear injection.

To generate the TRE-spastin mouse line, the spastin M85 sequence was obtained from the pEGFP-C1 spastin M85 plasmid (kind gift of Dr. Peter Baas). *SPAST* is the official name for the gene encoding spastin. First, the XhoI site within spastin M85 was mutated using site-directed mutagenesis (CTCGAG to CTAGAG) to generate a synonymous arginine mutation (amino acid 345) (CGA to AGA). An HA tag (TACCCATACGATGTTCCAGATTACGCT) was inserted at the N-terminus of the spastin M85 followed by a 4x glycine linker separating the HA tag and the start codon of spastin using PCR primers. SacII and BamHI sites were inserted on the 5' and 3' ends of the HA-spastin cassette, respectively. HA-spastin was PCR amplified, digested with SacII and BamHI, and inserted into pTre2 XhoI mut cut with the same. Doxycycline-dependent expression of the HA-spastin cassette was verified in cultured keratinocytes co-transfected with a K14-rtTA plasmid. The vector was linearized using XhoI and was used by the Duke Transgenic Core to generate transgenics via pronuclear injection.

To generate the K10-rtTA transgenic mouse line, the rtTA sequence was cloned behind the mouse keratin 10 promoter (BAC RP23-1D9) by the Duke Recombineering Core. Linearized DNA was used by the Duke Transgenic Core to generate transgenics via pronuclear injection.

Additional mouse lines used in this study were K14-rtTA (*Nguyen et al., 2006*), CMV-rtTA (Jackson labs), EMTB-GFP (*Lechler and Fuchs, 2007*) and TRE-H2B-GFP (*Tumbar et al., 2004*). For BrdU experiments, BrdU (10 mg/kg) was injected into adult mice, pregnant dams (for embryos) or neonatal pups. Animals were sacrificed one hour after BrdU injection for tissue dissection and processing.

### Mouse keratinocytes

All keratinocytes used were primary cells isolated from newborn mouse backskin. Full thickness backskin was treated with dispase II (Roche, Indianapolis, IN) overnight at 4°C and isolated epidermis was trypsin treated to generate single cells.

## Cornified envelope preparations

Cornified envelopes were isolated as previously described (*Sumigray et al., 2011*). Epidermis was isolated from P0 mice and boiled in 10 mM Tris (pH 7.4), 1% β-mercaptoethanol, and 1% SDS. Remaining corneocytes were pelleted and resuspended in PBS. Resuspended corneocytes were placed on slides for imaging.

## Transmission electron microscopy

Isolated P0 back skins were fixed in EM fix buffer (2% glutaraldehyde, 4% paraformaldehyde, 1 mM $CaCl_2$, 0.05M cacodylate pH 7.4) for one hour at room temperature and then were placed at 4°C. Subsequent fixation, embedding, and sectioning were performed as previously described (*Sumigray et al., 2011*).

## X-gal barrier assay

For the X-gal barrier assay, e18.5 embryos were placed into an X-gal solution (1 mg/ml X-gal, 1.3 mM $MgCl_2$, 100 mM $NaH_2PO_4$, 3 mM $K_3Fe[CN]_6$, 0.01% sodium-deoxycholate, 0.2% NP-40). After 5 hr, embryos were washed with PBS and photographed.

## Biotin barrier assay

To assess tight junction barrier function, P0 mouse pups were injected with 50 µl of 10 mg/ml NHS-biotin. After 30 min, pups were sacrificed and back skins were isolated and frozen in OCT. Biotin was detected using Streptavidin-FITC (Invitrogen).

## Cell culture

Stable wild-type keratinocytes were maintained in E low $Ca^{2+}$ media at 37°C. Plasmid transfection was performed using the Mirus transfection reagent (Mirus).

## Granular cell isolation and drug treatment

Granular cells were isolated from P0 mice. Epidermis was isolated by placing back skin into Dispase II (2.8 units/ml) for 1 hr at 37°C and 5% $CO_2$. Isolated epidermis was then placed into 0.25% Trypsin-EDTA with rotating for 1 hr to effectively isolate granular cells. Cells were washed with PBS, pelleted, and then resuspended in E low $Ca^{2+}$ media with either DMSO or 10 mg/ml nocodazole and placed at 37°C and 5% $CO_2$ for 1 hr. Cells were fixed in 4% PFA and stained with Rhodamine Ulex Europaeus Agglutinin one for visualization.

## Staining and antibodies

Tissue was embedded in OCT, frozen, and sectioned using a cryostat. Depending on the antibodies used, tissue sections were fixed either with room-temperature 4% PFA for seven minutes or ice-cold methanol for two minutes. Slides were washed with PBS + 0.2% Triton-X and blocked with BSA, NGS, and NDS before adding the primary antibody. The following primary antibodies were used in this study: rat anti-HA (11867423001, Sigma-Aldrich), rat anti-α-tubulin (sc-53029, Santa Cruz), rabbit anti-keratin 6 (PRB-169P, Covance), chicken anti-keratin 5/14 (generated in the Lechler lab), rabbit anti-keratin 10 (905401, Covance), rabbit anti-filaggrin (905801, Biolegend), rabbit anti-loricrin (kind gift from Colin Jamera), rat anti-BrdU (ab6326, Abcam), rabbit anti-active-caspase-3 (AF835, R and D systems), rat anti-β4 integrin (553745, BD Biosciences), rat anti-ECCD2 (kind gift from Colin Jamora), rabbit anti-keratin 1 (kind gift from Colin Jamora), mouse anti-desmoplakin (CBL173, Chemicon/Millipore), mouse anti-desmocollin-2/3 (clone 7G6, Santa Cruz), mouse anti-desmoglein-1 (610273, BD Biosciences), rabbit anti-phospho-myosin light chain 2 (Thr18/Ser19) (3674, Cell Signaling), rabbit anti-occludin (ab3172, Abcam), rabbit anti-ZO-1 (61–7300, Zymed/Invitrogen), rabbit anti-centrin1 (ab101332, Abcam), mouse anti-acetylated tubulin (T7451, Sigma-Aldrich), Rhodamine Ulex Europaeus Agglutinin 1 (RL-1062, Vector Laboratories). F-actin was visualized using fluorescently conjugated Phalloidin (A12379, Invitrogen and P1951, Sigma-Aldrich).

## Image acquisition

Images of the K14-rtTA; TRE-spastin keratinocytes and mice were acquired on a Zeiss Axio Imager microscope with Apotome attachment with the following objective lenses: 20x Plan-Apo 0.8 NA

lens, 40x Plan-Neofluar 1.3 NA oil lens, and 63x Plan-Apo 1.4 NA oil lens. Images were acquired using AxioVision software. All images of the K10-rtTA; TRE-spastin mice were acquired on a Zeiss Axio Imager microscope with Apotome 2 attachment and Axiocam 506 mono camera with the same objectives. When making intensity measurement comparisons, all images within one experiment were taken with identical exposure times.

Movies of EB1-GFP in CMV-rtTA; TRE-EB1 embryos were acquired on an Andor XD revolution spinning disc confocal microscope at 37°C and 5% $CO_2$ using a 60x Plan-Apo 1.2 NA water objective. Briefly, embryos (e11.5 (for basal), e15.5 (for spinous), or e17.5 (for granular)) were dissected and placed onto glass-bottom dishes (MatTek) in E media for imaging. EB1-GFP movies were acquired at 1 frame per second. Embryos were used for a maximum of one hour after isolation. Images on the spinning disc microscope were acquired using MetaMorph software. For imaging EB1-GFP in isolated, primary keratinocytes, epidermis was isolated from P0 back skin by overnight incubation in dispase (1 U/ml). Keratinocytes were isolated by incubation in 1:1 0.25% trypsin-EDTA: versene and plated onto glass-bottom dishes in E no $Ca^{2+}$ media supplemented with 0.5 mM $Ca^{2+}$ for 48 hr before imaging. Images of EB1-GFP in isolated, primary keratinocytes were acquired on a Leica DMI6000 microscope at 37°C and 5% $CO_2$ using a 63x Plan-Apo 1.4–0.6 NA oil objective. These images were acquired using SimplePCI software. For DMSO and nocodazole experiments, embryos were first screened for EB1-GFP expression on the Leica DMI6000 microscope, placed into E media with 10 µg/ml nocodazole or DMSO for 30 min at 37°C and then imaged on the spinning disc microscope.

### Image quantification

All image quantification was done using FIJI software. EB1-GFP comets were tracked manually by marking the starting and ending positions of an EB1 comet and noting the number of frames during which the comet was visible. All EB1 tracks were considered to be straight lines, and any that deviated sharply or curved were excluded from the analysis. For quantifying EB1 comet density, the number of EB1 comets was counted and the cell cytoplasm was manually outlined to calculate the cellular area. For quantification of α-tubulin intensity in K14-rtTA; TRE-spastin keratinocytes, each cell was outlined and the average HA and α-tubulin intensities were obtained for each cell. The mean intensities for each cell were then normalized to the maximum value in the picture (HA from spastin+ cells and α-tubulin from control cells). Aspect ratios for the differentiation-induced cell shape changes were calculated by tracing individual cells and using the measurement option in FIJI. Line-scan analysis of desmoplakin and desmocollin2/3 was performed by manually drawing 5 pixel-wide lines across cell-cell boundaries and calculating the mean fluorescence intensities at each point along the lines. Maxima were aligned and the ends were trimmed to yield the final line scan. All statistical analysis was performed using GraphPad Prism 5 software.

## Acknowledgements

We thank Peter Baas for reagents, Julie Underwood for care of the mice, and members of the Lechler Lab for valuable advice. In addition, we thank the Duke Transgenic Core for the generation of novel mouse lines. This work was supported by an NSF predoctoral grant to AM and by grants from the NIH (NIAMS and NIGMS), R01AR055926, R01AR067203 and R01GM111336 to TL.

## Additional information

### Funding

| Funder | Grant reference number | Author |
| --- | --- | --- |
| National Institute of General Medical Sciences | Research Grant | Terry Lechler |
| National Institute of Arthritis and Musculoskeletal and Skin Diseases | Research Grant | Terry Lechler |
| National Science Foundation | Graduate Student Fellowship | Andrew Muroyama |

The funders had no role in study design, data collection and interpretation, or the decision to submit the work for publication.

**Author contributions**
Andrew Muroyama, Conceptualization, Data curation, Formal analysis, Funding acquisition, Investigation, Methodology, Writing—original draft, Writing—review and editing; Terry Lechler, Conceptualization, Resources, Formal analysis, Supervision, Funding acquisition, Project administration, Writing—review and editing

**Author ORCIDs**
Andrew Muroyama (iD) http://orcid.org/0000-0003-0701-212X
Terry Lechler (iD) http://orcid.org/0000-0003-3901-7013

**Ethics**
Animal experimentation: All mouse studies were performed in accordance with our protocol (A147-15-05) approved by the Institutional Animal Care and Use Committee of Duke University.

**Decision letter and Author response**
Decision letter https://doi.org/10.7554/eLife.29834.023
Author response https://doi.org/10.7554/eLife.29834.024

## Additional files

**Supplementary files**
• Transparent reporting form
DOI: https://doi.org/10.7554/eLife.29834.022

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
