## [Decision Letter]

Thank you for submitting your article "A transgenic toolkit for visualizing and perturbing microtubules reveals unexpected functions in the epidermis" for consideration by *eLife*. Your article has been favorably evaluated by Fiona Watt (Senior Editor) and three reviewers, one of whom is a member of our Board of Reviewing Editors. The following individual involved in review of your submission has agreed to reveal her identity: Kathleen Janee Green (Reviewer #3).

The reviewers have discussed the reviews with one another and the Reviewing Editor has drafted this decision to help you prepare a revised submission.

Summary:

The strength of this paper is the development and characterization of these new models for studying MTs role in the epidermis. The tools will be of great use to the community and the observations made here are provocative and should be of broad interest to the *eLife* readership. They develop a practical approach to disrupt microtubules with temporal and tissue-specific control in intact mice. The fact that they get some quite surprising results indicates just how important it is to actually look at how quite familiar things like microtubule polymerization work in tissues. However, there are also some caveats associated with the authors' interpretations that would benefit from additional experimental analysis and/or discussion. Because of the complexity of the outcomes and interrelatedness of cytoskeletal regulatory pathways, the authors should be cautious not to overstate their conclusions.

Essential revisions:

1) Figure 4 should be moved to supplementary data (a supplement to a primary figure, per *eLife* policy).

2) The authors conclude in the Abstract and manuscript that the desmosome phenotype is non-cell autonomous based on data in Figure 6. Further quantification and images are needed to fully support this conclusion.

3) The characterization of entosis is interesting but without 3D shapes of the epidermis, how can the authors be confident of entotic cell analysis?

Major Issues:

4) The authors describe a defect in cell flattening in spastin positive cells, and conclude that MT are important for this process. They suggest that it is unlikely actin plays a role, based on previous observations from Arp3 and myosin II deficient mice. However, disruption of microtubules could result in the release and activation of factors such as GEFs and mDia, possibly leading to alterations in actin. Is it possible that GEF activity and/or mDia (for instance) are altered and contribute to the observed phenotypes?

5) The authors also do not adequately consider the potential role of loss of Dsg1 (shown in Figure 6) to the observed phenotype of hyperproliferation and epidermal thickening. It is interesting that these phenotypes somewhat resemble SPPK, which is sometimes linked to Dsg1 deficiency causing increased MAPK signaling.

6) The authors state that cell flattening is independent of desmosomes because spastin positive cells with normal cortical DP are unable to flatten. Related to this conclusion, cell flattening defect was analyzed in the K10-rtTA model, while the normal cortical DP was shown using the K14-rtTA line (Figure 6) and DP was significantly disturbed in the K10-rtTA line (Figure 6). In addition, it has been shown that Dsg1 deficiency results in changes in cell shape.

7) Also, there are caveats associated with relying on DP as an indicator of normal desmosome assembly/function as DP does not require MTs for its incorporation into junctions, but there could be interference with MT-dependent trafficking of specific forms of the desmosomal cadherins that are not appreciated in the analysis. So just because DP is present, doesn't mean that desmosomes exhibit their normal stage-specific structure/composition. Also, while the authors show tissue wide effects on desmosomes in the K10-rtTA model, they do not show this for the K14-rtTA model.

---

## [Author Response]

Essential revisions:1) Figure 4 should be moved to supplementary data (a supplement to a primary figure, per eLife policy).

Figure 4 has been moved to become a supplementary figure (it is now Figure 4—figure supplement 1).

2) The authors conclude in the Abstract and manuscript that the desmosome phenotype is non-cell autonomous based on data in Figure 6. Further quantification and images are needed to fully support this conclusion.

We have included in the revision more extensive quantifications to demonstrate this. A quantification of the desmoplakin intensity between pairs of cells that express spastin and are surrounded by wild type cells is now included to accompany the images (new panel Figure 5). This analysis was from K14-rtTA;TRE-spastin mice in which few suprabasal cells expressed spastin. This demonstrates that the cortical intensity of desmoplakin is equivalent between these mutant cell pairs compared to wild-type cell pairs. We also included a quantification of desmoplakin intensity from regions of these same mice where a high number of cells were spastin positive. These data (Figure 5—figure supplement 1) show the same phenotype as the K10-rtTA;TRE-spastin mice, a loss of cortical desmoplakin staining. We also include new images of Dsg1 staining (Figure 5—figure supplement 1) that show the same findings for this desmosomal cadherin. Together, these data are consistent with microtubules not having a cell-autonomous role in cortical localization of desmosomal proteins, but rather, that when enough cells have disrupted microtubules a non-cell autonomous response involves desmosome perturbation.

3) The characterization of entosis is interesting but without 3D shapes of the epidermis, how can the authors be confident of entotic cell analysis?

We have included a supplemental video through an entotic event (Supplementary Video 4). All images were collected as Z-stacks and in many cases we could optically section through entire cells and find them surrounded by other cells.

Major Issues:4) The authors describe a defect in cell flattening in spastin positive cells, and conclude that MT are important for this process. They suggest that it is unlikely actin plays a role, based on previous observations from Arp3 and myosin II deficient mice. However, disruption of microtubules could result in the release and activation of factors such as GEFs and mDia, possibly leading to alterations in actin. Is it possible that GEF activity and/or mDia (for instance) are altered and contribute to the observed phenotypes?

The text of our document stated that actomyosin contractility does not play a role in flattening (based largely on the fact that myosin II mutant mice undergo normal flattening). We did not address whether actin might play some role in this process. However, to make this clearer we have altered a sentence in the Discussion to read:

“One possibility is that microtubules influence cell shape indirectly through effects on other cytoskeletal elements, such as actin filaments, or on cell adhesions, such as desmosomes.”

While we cannot rule out a role for mDia, GEF activity would likely result in an increase in acto-myosin contractility, and we did not see any evidence for this in terms of either myosin II requirement or in levels of pMLC (a marker for active myosin II, data not shown).

5) The authors also do not adequately consider the potential role of loss of Dsg1 (shown in Figure 6) to the observed phenotype of hyperproliferation and epidermal thickening. It is interesting that these phenotypes somewhat resemble SPPK, which is sometimes linked to Dsg1 deficiency causing increased MAPK signaling.

We have included new images demonstrating that Dsg1 localization is also not affected by microtubule depolymerization in a cell-autonomous matter (Figure 5—figure supplement 1). As for the possibility that defects in desmosomes underlie the hyperproliferation – while SPPK is an excellent example in human skin disease, there is no data that desmosome mutants result in hyperproliferation in an embryonic context, and especially before barrier loss. While we cannot rule out this possibility, we think it is unlikely given that published work demonstrated that Dsg1 loss of function results in a differentiation defect, which we do not see in the spastin expressing mouse (rather, the basal cells are hyperproliferative, but differentiation occurs normally). Therefore, we aren’t convinced that the phenotypic similarity is sufficient to warrant this discussion.

6) The authors state that cell flattening is independent of desmosomes because spastin positive cells with normal cortical DP are unable to flatten. Related to this conclusion, cell flattening defect was analyzed in the K10-rtTA model, while the normal cortical DP was shown using the K14-rtTA line (Figure 6) and DP was significantly disturbed in the K10-rtTA line (Figure 6). In addition, it has been shown that Dsg1 deficiency results in changes in cell shape.

We now include data on the flattening defects in both the K14-rtTA and the K10-rtTA lines. Both show a consistent defect in granular cell shape that is cell autonomous (Figure 4 and Figure 4—figure supplement 3). We also include quantifications for DP cortical intensities in both K14-rtTA; TRE-spastin and K10-rtTA; TRE-spastin lines (Figure 5 and Figure 5—figure supplement 1). In addition, Figure 5—figure supplement 1 includes images showing the Dsg1 remains cortical in cells that are clearly rounded, suggesting that the rounding is unlikely due to loss of Dsg1 localization.

7) Also, there are caveats associated with relying on DP as an indicator of normal desmosome assembly/function as DP does not require MTs for its incorporation into junctions, but there could be interference with MT-dependent trafficking of specific forms of the desmosomal cadherins that are not appreciated in the analysis. So just because DP is present, doesn't mean that desmosomes exhibit their normal stage-specific structure/composition. Also, while the authors show tissue wide effects on desmosomes in the K10-rtTA model, they do not show this for the K14-rtTA model.

This is an excellent point and we agree that looking solely at DP may not be ideal, given the data on microtubule dependent transport of some desmosomal cadherins. We thus included images of Dsg1 as well (Figure 5 and Figure 5—figure supplement 1) and find the same results as with desmoplakin. In addition, the revised manuscript includes the analysis of both K10-rtTA and K14-rtTA mice, as discussed above.